# The Value of CXCL1, CXCL2, CXCL3, and CXCL8 as Potential Prognosis Markers in Cervical Cancer: Evidence of E6/E7 from HPV16 and 18 in Chemokines Regulation

**DOI:** 10.3390/biomedicines11102655

**Published:** 2023-09-28

**Authors:** Leonardo Fernandez-Avila, Aribert Maryosly Castro-Amaya, Andrea Molina-Pineda, Rodolfo Hernández-Gutiérrez, Luis Felipe Jave-Suarez, Adriana Aguilar-Lemarroy

**Affiliations:** 1Programa de Doctorado en Ciencias Biomédicas, Centro Universitario de Ciencias de la Salud (CUCS), Universidad de Guadalajara, Guadalajara 44340, Jalisco, Mexico; leonardofernavi@gmail.com; 2División de Inmunología, Centro de Investigación Biomédica de Occidente (CIBO), Instituto Mexicano del Seguro Social (IMSS), Guadalajara 44340, Jalisco, Mexico; aribert.castro@gmail.com; 3Centro de Investigación y Asistencia en Tecnología y Diseño del Estado de Jalisco, A.C., Guadalajara 44270, Jalisco, Mexico; andymopi@gmail.com (A.M.-P.); rhgutierrez@ciatej.mx (R.H.-G.); 4Consejo Nacional de Ciencia y Tecnología, CONAHCYT, Mexico City 03940, Mexico

**Keywords:** chemokines, HPV, E6, E7, cervical cancer, CXCL, prognosis marker, HaCaT cells, HeLa, SiHa, C33A

## Abstract

Cervical cancer (CC) is a serious global health issue, and it is well-known that HPV infection is the main etiological factor that triggers carcinogenesis. In cancer, chemokine ligands and receptors are involved in tumor cell growth, metastasis, leukocyte infiltration, and angiogenesis; however, information on the role played by E6/E7 of HPV16/18 in the modulation of chemokines is very limited. Therefore, this study aimed to determine whether chemokines are differentially expressed in CC-derived cell lines; if E6/E7 oncoproteins from HPV16 and 18 are capable of mediating chemokine expression, what is the expression profile of chemokines in tissues derived from CC and what is their impact on the overall survival of patients with this pathology? For this purpose, RNA sequencing and real-time PCR were performed on SiHa, HeLa, and C33A tumorigenic cell lines, on the non-tumorigenic HaCaT cells, and the *E6/E7* HPV-transduced HaCaT cell models. Furthermore, chemokine expression and survival analysis were executed on 304 CC and 22 normal tissue samples from The Cancer Genome Atlas (TCGA) repository. The results demonstrate that *CXCL1*, *CXCL2*, *CXCL3*, and *CXCL8* are regulated by *E6/E7* of HPV16 and 18, are overexpressed in CC biopsies, and that their higher expression is related to a worse prognostic survival.

## 1. Introduction

Cervical cancer (CC) is recognized as the fourth most common female carcinoma worldwide, and its frequency is notably higher in many low- and middle-income countries (LMICs) [1]. In 2020, 604,127 new cases and 341,831 deaths were reported globally [2]; approximately, 84% of the new cases and 90% of the deaths occurred in LMICs [3]. In fact, in Mexico, about 7900 new cases are diagnosed annually, which situates cervical cancer in the third and second place in incidence and mortality, respectively, in pathologies associated with cancer in women [4].

The cervix can be divided into two mature epithelia, the squamous epithelium (ectocervix) and glandular epithelium (endocervix), and the region where both types of cells converge is defined as the transformation zone, which is highly susceptible to human papillomavirus (HPV) infection and tumor development [5].

Unfortunately, women with early-stage HPV infections are often asymptomatic, and clinical indications of tissue damage typically only appear in the advanced stages [6]. HPV is the most common sexually transmissible infection worldwide as a sexually active person will get infected at least once in their lifetime [7]. HPV belongs to the *Papillomaviridae* family of non-enveloped DNA viruses of which roughly 450 genotypes have been identified [8], and fifteen are considered carcinogenic to humans (high-risk HPV) [9]. Persistent infection of high-risk HPV is a necessary cause of CC as it is associated with nearly 99% of CC cases [10,11]; among the high-risk genotypes, HPV16 and HPV18 are identified as the most prevalent due to a correlation of their presence with 70% of CC cases [12].

The carcinogenic potential of high-risk HPVs is derived from the early expression of two dynamic and versatile oncoproteins, E6 and E7, which manipulate the host cell to transform itself and establish a proper tumor niche [13]. Indeed, cervical cancer persistence requires the constant presence of E6 and E7 oncoprotein activity [14]. Another report describes that cancer cells could go into senescence or apoptosis in the absence of E6/E7 functions [15]. The most important functions of these oncoproteins are the inactivation of the tumor suppressor proteins p53 and pRB [16,17]; the activation of hTERT (human telomerase reverse transcriptase) associated with cell immortalization [18]; the loss of the cell polarity and cell–cell unions due to the linkage of E6 and the membrane-associated guanylate kinase proteins (E6/MAGUK) [19,20] and the suppression effect of E7 on the antigen processing pathway through the transporter associated with antigen protein 1 (TAP1) [20,21]. In summary, both oncoproteins act cooperatively in the disease; E7 controls early carcinogenesis while E6 hastens progression towards malignancy [22,23]. On the other hand, E6 and E7 oncoproteins can regulate the expression of immune mediators in host cells [24], such as cytokines and chemokines, to promote immune cell recruitment, modulation of the tumor microenvironment [25], angiogenesis, inflammation [26], and tumor cell proliferation [27].

Chemokines are a group of nearly 50 small proteins with fundamental chemotactic functions, such as adhesion, migration, localization, and cell–cell interactions by binding with their G protein-coupled receptors [28]. A common subgroup of chemokines is secreted without stimuli, called homeostatic chemokines, and are related to immune surveillance [29]; however, under particular circumstances, i.e., cancer, inflammatory chemokines released will act as chemotactic agents for circulatory leukocytes at the highest concentration site [30].

In cancer, tumors are characterized by wide cell heterogenicity that communicate to each other in a complex signaling network [31]. This crosstalk involves the expression both chemokine ligands and receptors by tumor cells, stromal cells (fibroblasts and endothelial cells), and infiltrating immune cells. In the tumor niche, transformed cells have the capability to develop new features to produce growth-promoting chemokines and increase the expression of chemokine receptors [32], creating a feedback loop that can cause more cancer cells to divide and hijack multiple processes, such as: tumor cell proliferation [33,34,35], metastasis [36,37], leukocyte infiltration [38,39,40], and angiogenesis [41].

In relation to pre-cancerous cervical lesions, Bhatia et al. tested the expression of 31 chemokines on cervical liquid-based cytology samples from cervical intraepithelial neoplasia (CIN grades 1–3 related to tumor progression); their findings suggest a change in the chemokine panel related to the worsening of CC [42]. Their research precedes the beginning of the formation of a biomarkers panel based on chemokines in the interest of ascertaining the CC prognosis. Despite the available knowledge, whether there is a relation between the expression E6 and E7 oncoproteins and the increase in chemokines is still inconclusive.

The present work aimed to evaluate the chemokine profile in CC-derived cells and tissues, whether the expression of chemokines is mediated by E6 and E7 from HPV16 and 18 oncoproteins, and to assess the correlation of the expressed chemokines with the overall survival of CC patients. All the previous aims are in the pursuit of identifying potential prognostic biomarkers for earlier stages of the disease.

## 2. Materials and Methods

### 2.1. Cell Culture

Non-tumorigenic keratinocytes (HaCaT), SiHa, HeLa, and C33A cell lines were obtained from the CIBO-IMSS cell bank repository. All cell lines were previously authenticated using the Multiplex Human Cell Line Authentication Test at Multiplexion GmbH (Friedrichshafen, Germany). The cells were cultured in Dulbecco’s Modified Eagle Medium (DMEM) supplemented with GlutaMAX™, D-glucose (1 g/L) (Cat. No. 10567-014 Gibco, Thermo Fisher Scientific, Waltham, MA, USA), penicillin (100 U/mL), streptomycin (100 μg/mL), sodium pyruvate (110 mg/L), and 10% fetal bovine serum (FBS). Cells were grown at 37 °C in a 5% CO_2_ atmosphere.

HaCaT models expressing the oncogenes *E6/E7* from HPV16 and HPV18 were previously established as described by Castro-Amaya, AM. et. al., 2022 [27]. HaCaT-derived models were named as follows: HaCaT with empty vector (HaCaT pLVX), HaCaT cells transduced with E6/E7 from HPV16 (HaCaT E6/E7 HPV16), and HaCaT cells transduced with E6/E7 from HPV18 (HaCaT E6/E7 HPV18). Expression of E6/E7 was confirmed via real-time PCR, the amplicons were run via electrophoresis on 2% agarose gels (as shown in Appendix A).

### 2.2. Quantitative PCR

Total RNA was isolated from approximately 10^7^ cells using the Quick-RNA miniprep plus kit (Cat. No. R1058, Zymo Research, Irvine, CA, USA) according to manufacturers’ instructions. The cDNA synthesis was performed with the Transcriptor First Strand cDNA Synthesis Kit (Cat. No. 04 379 012 001, Roche Diagnostics, Basel, Switzerland). Quantitative PCR (qPCR) was performed on the LightCycler 2.0 platform (Roche Diagnostics, Basel, Switzerland) using the LightCycler FastStart DNA Master plus SYBR Green I Kit (Cat. No. 03515869001, Roche Diagnostics, Basel, Switzerland). To normalize the expression data, the *RPS18* and *RPLP0* were taken as reference genes. The results were examined with the LightCycler Software 4.1. Relative expression was calculated using the 2-ΔΔCp algorithm and transformed to Log2FC. Primers were designed using the NCBI-Primer-BLAST application (https://www.ncbi.nlm.nih.gov/tools/primer-blast/, accessed on 9 December 2021); primers’ set sequences are listed in Appendix A.

### 2.3. RNAseq

RNA extracted from two independent samples was shipped for the Next-generation RNAseq sequencing service from the Novogene Corporation Inc. (Sacramento, CA, USA), who perform library preparation and sequencing in a NovaSeq 6000 (Illumina, San Diego, CA, USA). All datasets were deposited in the GeoData repository with the accession numbers GSE241703 and GSE241704.

### 2.4. Bioinformatics

The open-source platforms RStudio (v2023.06.1+524) and Galaxy (https://www.usegalaxy.org, accessed on 10 July 2023) were used to analyze the Illumina raw data. The quality of the reads was performed through the FastQC tool (v0.12.1) [43]. All the reads were mapped using the Aling aligner of the Rsubread package (v2.14.2) [44] and the Human Genome Reference (GRCh38.p14 v43) [45]. The resulting outputs were BAM files, of which the number of reads per gene were determined via the featureCounts tool (v2.0.3) [46]. Count tables generated via the featureCounts tool were used to perform a differential expression analysis using the DESeq2 tool (v2.11.40.8) [47]. The differential gene expression measurements were normalized using DESeq2’s median of ratios (median of ratio of gene counts relative to geometric mean per gene) method. All chemokine genes with an adjusted *p*-value (*adj p*) ≤ 0.05 and fold change (Log2FC) ≥ 1 were selected as differentially expressed genes (DEGs).

### 2.5. Expression Profile

The expression profile of the selected chemokines (*CCL2*, *CCL28*, *CXCL1*, *CXCL2*, *CXCL3*, *CXCL6*, *CXCL8*, *CXCL10*, and *CXCL11*) in CC samples’ expression analysis was conducted through the OncoDB platform (https://oncodb.org/index.html, accessed on 19 June 2023) [48]. This was based on “The Cancer Genome Atlas” (TCGA) database (https://www.cancer.gov/tcga, accessed on 19 June 2023), where the data of cervical squamous cell carcinoma (CESC) together with normal tissue were extracted (n = 304 and n = 22, respectively) and compared.

### 2.6. Survival Analysis

The correlation between chemokine panel expression and overall survival (OS) was performed with the Kaplan–Meier Plotter platform (http://kmplot.com/analysis/index.php?p=service&cancer=pancancer_rnaseq, accessed on 25 June 2023) [49]. This was based on 304 CESC samples from the TCGA database, with a follow-up threshold of 60 months.

### 2.7. Statistical Analyses

All experiments were completed with at least twofold independent replicates. For qPCR analysis to determine the differences between groups, an analysis of variance (ANOVA) and Dunnett’s test were performed. Regarding the expression profile, the Student’s *t*-test with Welch’s correction were selected to demonstrate the disparity between the control and CC samples. All the output results from the statistical instruments previously mentioned were expressed as mean ± standard deviation (SD), and *p*-value (*p*) ≤ 0.05 was designated as statistically significant, furthermore, all data were processed in GraphPad Prism v8.0.1 software. Concerning differential expression analysis, all the genes investigated for each cellular model and cell line were accomplished with the following criteria: adjusted *p*-value (*adj p*) *≤* 0.05 and fold change value of at least 1 compared to the control group. For this data, the statistics were acquired through the Galaxy platform.

## 3. Results

### 3.1. Differential Chemokines Expression between Cervical Cancer-Derived Cell Lines and Non-Tumorigenic Keratinocytes

To ascertain if there is differential chemokines expression between CC-derived cell lines and non-tumorigenic keratinocytes, we assessed the RNA sequencing of SiHa (HPV16^+)^, HeLa (HPV18^+)^, C33A (HPV^−^) and as control the non-tumorigenic HaCaT cells (HPV^−^), taking HaCaT cells as the calibrator, as depicted in Figure 1a. The results show 18 chemokines were differentially expressed with statistical significance (*adj p* ≤ 0.05); those that were determined to be overexpressed in both HPV^+^ cell lines were: *CCL20*, *CXCL2*, *CXCL3*, *CXCL10*, and *CXCL11*; while those observed as strongly downregulated were: *CCL2*, *CCL17*, *CCL22*, *CXCL1*, *CXCL6*, *CXCL8*, and *CXCL16.* In that sense, the fold-changes observed were from 6.45 to −11.03. On the other hand, the C33A cell line presents a predominant tendency to downregulation, and the overexpressed chemokines in the SiHa and HeLa cell lines are shown in C33A cells as non-differentially expressed. All this allows us to infer that HPV’s presence in CC-derived cell lines could modify the chemokine expression.

### 3.2. E6/E7 HPV16 and 18 Modulate the Chemokines Expression

To determine if the mere presence of the *E6/E7* oncogenes of HPV16 and 18 can modulate the expression of chemokines, the non-tumorigenic keratinocytes (HaCaT cells) were transduced with *E6/E7* from each HPV. They were used as cell models to evaluate the impact of the oncoproteins on chemokine expression. The cell models were named as follows: HaCaT pLVX (transduced with empty vector), HaCaT *E6/E7* HPV16, and HaCaT *E6/E7* HPV18. RNAseq analyses using HaCaT pLVX as the calibrator identified 14 differentially regulated chemokines of statistical significance (*adj p* ≤ 0.05), as shown in Figure 1b. As opposed to the previous SiHa, HeLa, and C33A findings, most of the chemokines have an overexpression tendency, with fold-change values ranging from 1.18 to 6.63. 

Remarkably, the *E6/E7* HPV18 model triggers the overexpression of a higher quantity and rate of chemokines, albeit *CCL2*, *CCL28*, *CXCL1*, *CXCL2*, *CXCL5*, *CXCL6*, *CXCL8*, and *CXCL10* are highlighted since they maintain the same trend in both models. In addition, *CXCL3*, *CXCL11*, *CXCL14*, and *CXCL17* were significantly increased only in HaCaT *E6/E7* HPV18.

To confirm previous findings, we decided to validate the data utilizing quantitative real-time PCR (qPCR), using HaCaT pLVX cells as the calibrator (set as 0). Normalization was performed using *RPLP0* and *RPS18* as reference genes. Primers of all genes, whose expression was observed to increase in the HaCaT *E6/E7* HPV18 model, were designed and synthesized; however, not all of them were amplified adequately and specifically. Therefore, the genes in which their expression could be validated with a statistically significant differential expression in both cell models were *CCL2*, *CCL28*, *CXCL2*, *CXCL3*, *CXCL6*, CXCL8, *CXCL10*, and *CXCL11*. *CXCL1* exhibited a statistically significant increase only in HaCaT *E6/E7* HPV18 cells, see Figure 2.

### 3.3. High Expression of Chemokines in Cervical Cancer Tissues

Once the expression of the various chemokines was evaluated in the HaCaT models, we were interested in elucidating whether the expression of the genes modulated by HPV16 or 18 was also reflected in biopsies derived from cervical cancer.

For this purpose, cervical squamous cell carcinoma (CESC) and normal cervical tissue data were downloaded from TCGA (n = 304 and n = 22, respectively) and compared to illustrate the local cervical environment better and show the relevance of the chemokines in that pathology.

As depicted in Figure 3, except for *CCL2*, all chemokines in CC maintained an extremely high expression with statistical significance, in contrast to normal tissues. The Log2fold-change observed in each of them were as follows: *CCL28* (0.69), *CXCL1* (4.84), *CXCL2* (1.58), *CXCL3* (2.41), *CXCL6* (2.14), *CXCL8* (5.22), *CXCL10* (5.72), and *CXCL11* (3.94).

### 3.4. Prognostic Outcome Linkage with Chemokines Expression in Cervical Cancer Patients

To determine whether the chemokine expression was linked to survival, Kaplan–Meier analyzes were performed on the same samples included in the TCGA-CESC database; the patient’s samples were divided into low- and high-expression groups and the ratio of death probability (hazard ratio—HR), and the statistical significance values were retrieved from a medium-term of 60 months. As can be observed in Figure 4, the analyzed data showed that a worse survival was linked to a high expression of *CXCL1* (HR = 2.52), *CXCL2* (HR = 2.57), *CXCL3* (HR = 2.49), and *CXCL8* (HR = 3.2); patients who show a high expression of each of these genes could have a two to three times higher probability of presenting with deterioration or complications during the first 60 months of the course of the pathology. On the other hand, a better survival was detected in the group of patients with a higher expression of *CXCL10*. Finally, non-statistically significant values were noted for *CCL2*, *CCL28*, *CXCL6*, and *CXCL11*.

## 4. Discussion

A crucial component in CC biology is the tumor microenvironment (TME) which functions as bidirectional communication among stromal and tumoral cells, leading to tumor initiation, progression, metastasis, and therapeutic response. Paracrine molecules, such as cytokines and chemokines, drive the crosstalk in the TME. Chemokines regulate various tumor cell properties, including stem-like cell features, proliferation, invasiveness, and neoangiogenesis in stromal cells [50,51]. Furthermore, they control immune cell biology characteristics such as activation, recruitment, phenotype, and function by controlling their location and interactions in lymphoid tissues and the TME [52,53]. The role of chemokines is complex, as their expression is regulated by immune cells, stromal cells, and tumor cells, contributing to both anti- and pro-tumorigenic immune responses, relying on various factors [54].

On the other hand, as previously mentioned, the most critical risk factor for women developing CC is persistent HPV infections, mainly by high-risk genotypes. The role of HPV oncoproteins like E6 and E7 in carcinogenesis involves intricate pathways that modulate a significant group of molecules [55], such as chemokines [27,56]. In our work, we investigated the expression of chemokines in CC-derived cell lines, the impact of E6/E7 oncoproteins in their expression level, and their effect on overall survival.

To make it easier to discuss the results, we made a summary table of all the results, shown in Appendix A. Our results revealed that expression of *CXCL1*, *CXCL2*, *CXCL3*, and *CXCL8* increases in the presence of *E6/E7* of HPV16 and 18, are overexpressed in CC biopsies, and that their higher expression is related to a worse prognostic survival in cervical cancer. In the literature, few studies evaluate the effect of HPV oncoproteins, specifically on chemokines. A study very similar to ours is the one carried out by Dai et al. [57], in which they also obtained models of HaCaT cells that overexpress *E6/E7* from HPV16, however, they used a different vector. In that work, in the Appendix A they reported an overexpression of *CXCL1*, but an under-expression of *CXCL2* in their *E6/E7* HPV16 expression HaCaT model.

In regard to *CXCL1*, it has been reported with an increased expression in cervical tumors [58], as well as in the serum of CC patients [59]. Another significant finding is its impact on survival in patients with this pathology, as its overexpression correlates with a worse prognosis. This could be due to the fact that *CXCL1* expression is related to increased proliferation and immortalization [60], the migration of carcinogenic cells [61], and angiogenic activation [62], as well as more severe cancer progression [63,64]. 

Concerning *CXCL2*, its participation has been constantly related to different types of cancer such as prostate, colon, and bladder [65,66,67], allowing a better understanding of its role in the tumor microenvironment. According to Jiang et al., *E6* splicing increases *CXCL2* expression and is linked to the infiltration of common lymphoid progenitor, neutrophils, and T γδ lymphocytes, while dendritic and CD8+ cells are absent [68]. In addition, *CXCL2* expression has been positively correlated with lymph node metastasis [69,70].

Regarding *CXCL3*, their roles reported in CC are related to the induction of proliferation and migration through the activation of ERK signaling pathway genes [71]. In addition, CXCL3 promotes cell growth and movement via autocrine/paracrine mechanisms involving Jak-STAT and MAPK signaling pathways in prostate [72] and oral squamous cell carcinoma [73].

In the case of CXCL8, in agreement with our results, analysis of cervical mucus in patients with CIN or CC revealed that HPV positivity was associated with increased expression of *CXCL8* [74], and low levels were linked with longer overall survival in CC patients [58]. Furthermore, Yan et al. demonstrated that its mRNA expression and protein concentration was exclusively from cervical cancer cell lines compared to cervical epithelial cell lines, and its protein levels significantly correlate to clinical stage, metastasis, histological type, and grade [75]. Moreover, through an autocrine mechanism, CXCL8 increases cell viability and proliferation under low-oxygen conditions in HeLa and SiHa cells [76].

In relation to *CXCL10*, we found it highly expressed in all the models analyzed; in agreement with that observation, similar results were obtained in tumoral tissue, where protein levels were higher at the tumor site than in the adjacent tissue [58]. On the other hand, its high expression was correlated with a better prognosis in overall survival analysis. These findings could be explained by Zhao et al.’s report of the chemokine’s relation to the inhibition of angiogenesis, a decrease in proliferation, S phase transition, and stimulation of apoptosis combined with radiotherapy [77]. Additionally, *CXCL10* has been related to the exosomal expression of PD-L1 by fibroblasts; the upregulated expression of *PD-L1* was observed only in HPV-positive patients via microarray analysis [78]; this mechanism could contribute to viral immune escape and carcinogenesis.

Expression of *CXCL6* was observed to be highly expressed in the *E6/E7* HPV16 and 18 HaCaT models, as also reported by Dai et al. [57] In addition, its expression was observed with the same tendency in and CC tissues. The biological role of CXCL6 in CC is to promote tumor growth, metastasis, and aggressiveness [79]. Moreover, the co-expression of a glycosylating enzyme, *GALNT2*, and *CXCL6* indicates poor overall survival of the disease [80]. 

*CXCL11* is usually associated with cancer promotion, including proliferation, migration, angiogenesis, and T-cell infiltration [81,82,83]. Higher levels of CXCL11 were found in tumoral tissue compared to adjacent tissue in CC samples [58]. Moreover, EphA2 mediated a relationship between high levels of CXCL11 and PD-L1, as reported by Zhao et al. [84]. However, studies have also shown that low levels of this chemokine can be associated with the severity of cervical lesions [85] and high-risk scores for cervical, ovarian, and urogenital cancers [86]. 

CCL2 plays a vital role in activating and recruiting macrophages in cancer progression. As a result, it is relevant in multiple types of cancer and has been associated with poor prognosis in digestive, head and neck, respiratory, and urogenital cancers [87]. In CC, high expression levels of *CCL2* have been linked to proliferation, migration, epithelial-mesenchymal transition, and perineural invasion [88]. 

The results found for *CCL28* show increased expression in the E6/E7 HaCaT models but only slightly increased in cervical biopsies, so its impact on the survival of patients with CC is not significant. However, Yan, J. et al. demonstrated that CCL28 has the ability to regulate the progression of pancreatic ductal adenocarcinoma through its autocrine activity [89]. It is also involved in the migration of pancreatic tumor cells, Treg cells, and pancreatic stellate cells, associating the *CCL28* overexpression with a worse prognosis in these patients [90]. In CC, *CCL28* expression was identified as being higher in high-grade CIN compared to HPV-negative samples and those with normal cytology [42].

As previously mentioned, chemokines are highly involved in the carcinogenic processes of different types of cancer in such a way that our study allows us to approach the plethora that triggers the presence of the E6 and E7 oncoproteins of HPV16 and 18. It is important to highlight that one limitation of the study was that the expression of the oncogenes in the *E6/E7* HPV16 HaCaT model was not as efficient as that observed in the *E6/E7* HPV18 HaCaT model, which could explain the differences observed in chemokine modulation by the oncoproteins of both HPV genotypes.

## 5. Conclusions

This study revealed that the presence of HPV16 and 18 oncogenes significantly alter the expression of *CCL28*, *CXCL1*, *CXCL2*, *CXCL3*, *CXCL6*, *CXCL8*, *CXCL10*, and *CXCL11*; overexpression of those chemokines was also confirmed in cervical cancer biopsies. Furthermore, the overexpression of some of them, *CXCL1*, *CXCL2*, *CXCL3*, and *CXCL8*, was linked to an unfavorable prognosis; meanwhile, expression of *CXCL10* has the opposite effect. This evidence gives us an approach to the usefulness of a chemokine’s levels on a patient’s fate. The directions we have to continue in with this research are to determine the expression of each of these genes at the protein level in the HaCaT-transduced cell models and cervical cancer-derived cell lines, as well as in samples of cervical swabs from women with low-grade cervical lesions, who will be undergoing colposcopic follow-up every six months, to assess their potential as a marker for predicting progression.

## Figures and Tables

**Figure 1 biomedicines-11-02655-f001:**
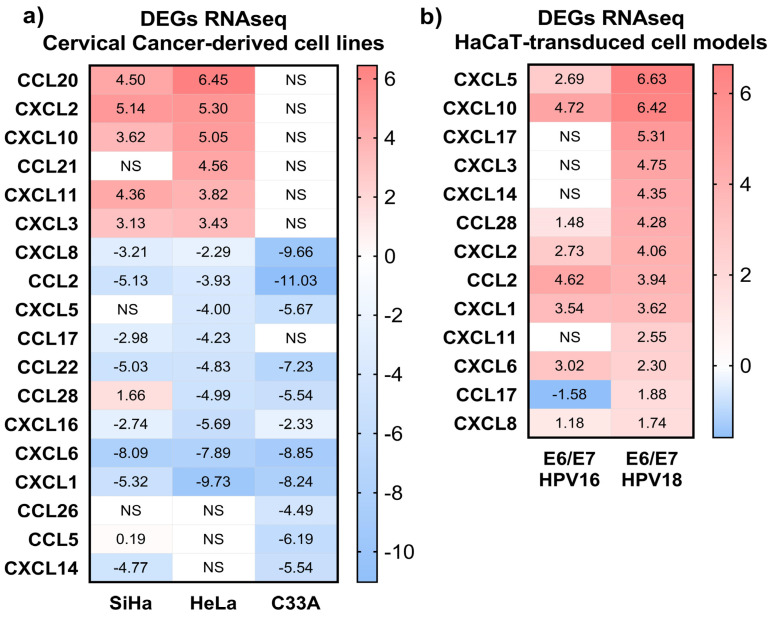
**Differentially expressed genes in cervical cancer-derived cell lines and in *E6/E7* HPV16 and 18-transduced HaCaT cells**. (**a**) The heat map shows the fold-changes (log2FC) of the modulated chemokine genes in each SiHa, HeLa, and C33A cell line. Each group has been contrasted against the expression of HaCaT, taking it as the calibrator. (**b**) The heat map depicts the fold-changes (log2FC) of the chemokines identified in the HaCaT models *E6/E7* HPV16 and *E6/E7* HPV18. Each group has been compared against the expression of HaCaT pLVX (empty vector), taking it as the calibrator. The color of the cells in both heat maps ranges from blue to red according to their expression, with statistical significance (*adj p* ≤ 0.05). The white cells depicted with NS show a non-significant differential expression. DEGs: differentially expressed genes.

**Figure 2 biomedicines-11-02655-f002:**
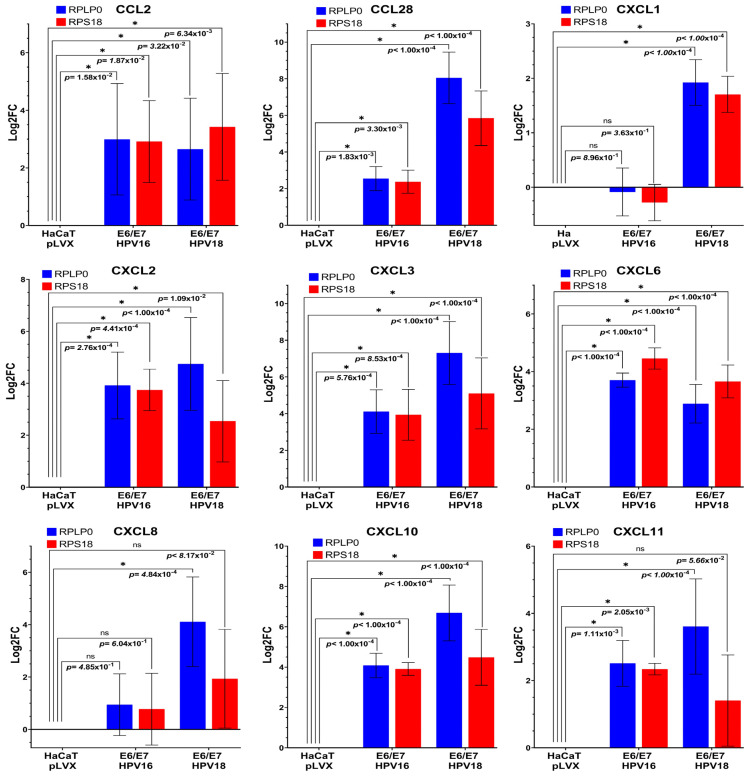
**Real-time PCR assays were performed on the HaCaT cell models.** Graphs show the relative expression (Log2FC) determined in each of the chemokines measured in HaCaT pLVX (empty vector), HaCaT *E6/E7* HPV16, and HaCaT *E6/E7* HPV18) employing the LightCycler 2.0 System; *RPLP0* and *RPS18* were used as reference genes. The expression of the chemokines in HaCaT pLVX was used as a calibrator. Statistical significance (*p*-value < 0.05 obtained through Dunnet’s test) is shown with an asterisk (*), (ns) indicates statistically non-significant.

**Figure 3 biomedicines-11-02655-f003:**
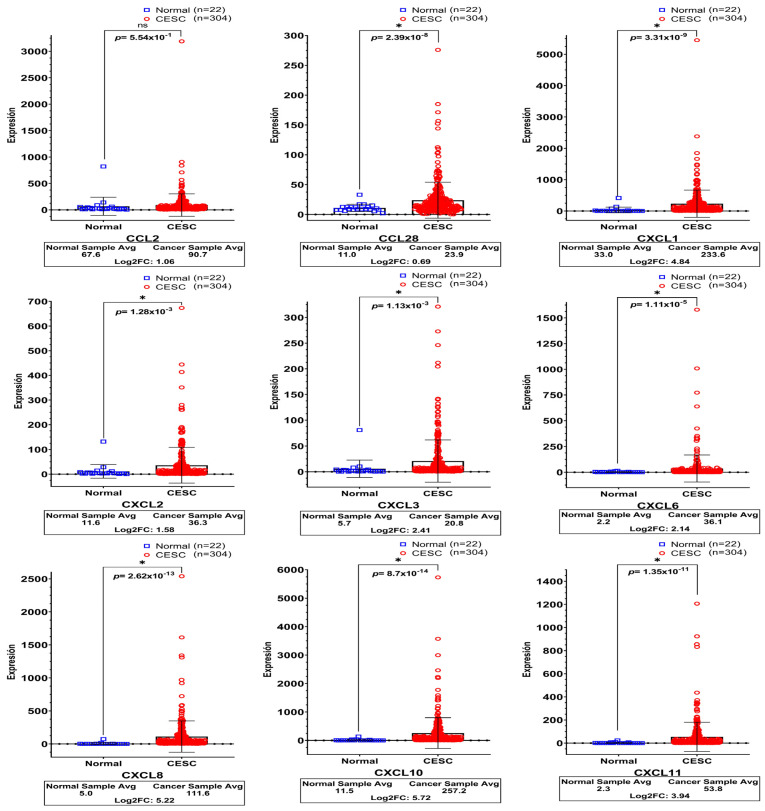
**Relative Expression Profiles of Chemokines Determined in Cervical Cancer Tissues.** The graphs exhibit the relative expression of each chemokine determined in 304 samples with cervical squamous cell carcinoma (CESC) versus 22 healthy tissues (Normal). The database was downloaded from The Cancer Genome Atlas Program (TCGA) repository through the OncoDB platform. (*) indicates statistical significance with a *p*-value < 0.05 obtained through Student’s *t*-test with Welch’s correction.

**Figure 4 biomedicines-11-02655-f004:**
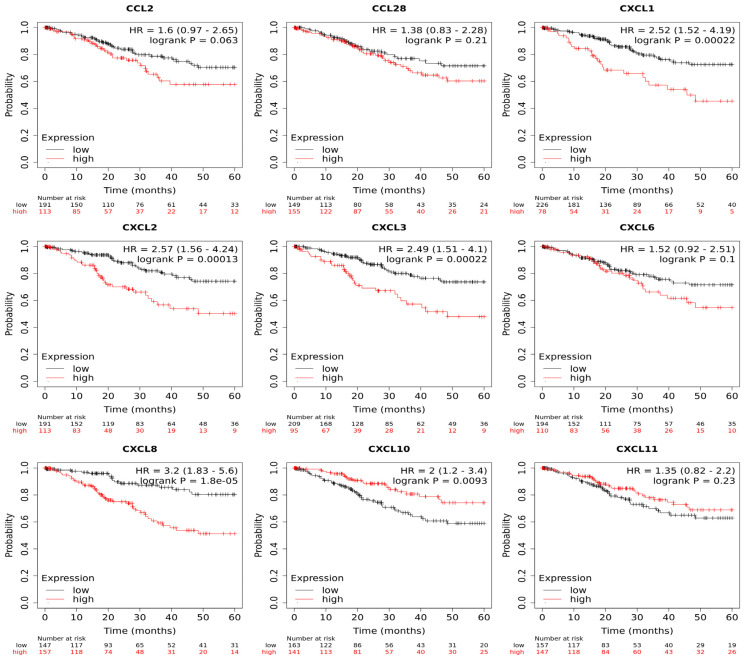
**Overall survival probability analysis of each of the Chemokines Determined in Cervical Cancer.** The plots illustrate the results derived from the “Kaplan-Meier Plotter” platform utilizing the data from TCGA repository. Survival curves represent the relationship between the probability of survival (displayed during the 60 months) and low or high expression. HR: hazard ratio; Logrank *p*: log rank *p*-value.

## Data Availability

The RNAseq datasets presented in this study were deposited on 24 August 2023 into the Gene Expression Omnibus (GEO) database repository (https://www.ncbi.nlm.nih.gov/geo, accessed on 24 August 2023), with the GEO accession numbers GSE241703 (Cervical cancer-derived cell lines), and GSE241704 (E6/E7 HPV16 and 18 HaCaT models). Datasets will be openly available on 15 February 2024.

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
