# Peer review of "The Value of CXCL1, CXCL2, CXCL3, and CXCL8 as Potential Prognosis Markers in Cervical Cancer: Evidence of E6/E7 from HPV16 and 18 in Chemokines Regulation"

_biomedicines, 2023, doi:10.3390/biomedicines11102655_

Round 1
Reviewer 1 Report
Comments:
The manuscript describes " The value of CXCL1, CXCL2, CXCL3, and CXCL8 as potential prognosis markers in cervical cancer: evidence of E6/E7 HPV16 and 18 chemokines regulation". This paper shows the chemokine profile in CC-derived cells and tissues, whether the expression of chemokines is mediated by E6 and E7 from HPV16 and 18 oncoproteins, and assesses the expressed chemokines with the overall survival of CC patients, but several points need to be clarified.
Comment:
1. According to different statistics and studies, more than 50 different chemical factors have been discovered. In this paper, the results show 18 chemokines differentially expressed with statistical significance. Authors are advised to list chemokines in your study between CC-derived cell lines and non-tumorigenic keratinocytes in Figure 1a.
2. Table 1. Primer sets sequences used in qPCR. This table shall be moved to a supplementary file.
3. Statistically significantly differentially expressed in both cell models were CCL2, CCL28, CXCL2, CXCL3, CXCL6, CXCL8, CXCL10, and CXCL11. CXCL1 only showed a significant increase in HaCaT E6/E7 HPV18 cells. Can the authors further confirm the difference in its protein expression?
Author Response
RESPONSE TO REVIEWERS' COMMENTS
Thank you for all the comments; they allowed us to make a better version of the manuscript. All changes suggested by the Editor and reviewers were done, as described below:
Reviewer 1:
Comments: The manuscript describes " The value of CXCL1, CXCL2, CXCL3, and CXCL8 as potential prognosis markers in cervical cancer: evidence of E6/E7 HPV16 and 18 chemokines regulation". This paper shows the chemokine profile in CC-derived cells and tissues, whether the expression of chemokines is mediated by E6 and E7 from HPV16 and 18 oncoproteins, and assesses the expressed chemokines with the overall survival of CC patients, but several points need to be clarified.
Comment: 1. According to different statistics and studies, more than 50 different chemical factors have been discovered. In this paper, the results show 18 chemokines differentially expressed with statistical significance. Authors are advised to list chemokines in your study between CC-derived cell lines and non-tumorigenic keratinocytes in Figure 1a.
R: We agree with your point; there are more than 50 different chemokines reported in the literature. However, in this study, we are showing only the chemokines whose differential expression was ≥ 1 (Log2FC), with a statistical significance ≤ 0.05 (adj p-value) in at least one of the CC-derived cell lines. We consider omitting those that do not comply with the selected parameters to improve the clarity of the result since all these omitted chemokines do not show any relevant value, and the value is depicted as "NS" (Non-significant differential expression).
- Table 1. Primer sets sequences used in qPCR. This table shall be moved to a supplementary file.
R: As suggested, Table 1. Primer sets sequences used in qPCR, has been moved to the supplementary section, named as follows: Table S1.
- Statistically significantly differentially expressed in both cell models were CCL2, CCL28, CXCL2, CXCL3, CXCL6, CXCL8, CXCL10, and CXCL11. CXCL1 only showed a significant increase in HaCaT E6/E7 HPV18 cells. Can the authors further confirm the difference in its protein expression?
R: Confirming the differences at the protein level is necessary to improve our findings, and we have already considered this. Indeed, we are designing the project's second phase, which involves confirming all the expression differences at the protein level, not only in the HaCaT-transduced cell models and cervical cancer-derived cell lines but also in cervical swab samples taken from women with low-grade cervical lesions. These patients will undergo colposcopic follow-up every six months to assess the potential of these genes as a marker for predicting progression. In addition, we only have ten days to resubmit the new version of the manuscript, and it is not feasible to carry out the requested experiments within this timeframe.

Reviewer 2 Report
Dear Editor,
I reviewed the manuscript by Fernández-Avila et al., entitled " The value of CXCL1, CXCL2, CXCL3, and CXCL8 as potential 2 prognosis markers in cervical cancer: evidence of E6/E7 HPV16 3 and 18 chemokines regulation"
In the present study, the authors indicated that CXCL1, CXCL2, 27 CXCL3, and CXCL8 are regulated by E6/E7 of HPV 16 and 18 in addition to be overexpressed in Cervical Carcinoma biopsies, and confirmed that the higher expression of the these chemokines is related to a worse prognostic survival.
The maunscript is well writen and the data are represent in best form and the support the main context of the study. In addition , this study will be interest to the reader of the journal. According the manuscript can be published in the present form.
Many thanks
Author Response
Reviewer 2:
I reviewed the manuscript by Fernández-Avila et al., entitled " The value of CXCL1, CXCL2, CXCL3, and CXCL8 as potential 2 prognosis markers in cervical cancer: evidence of E6/E7 HPV16 3 and 18 chemokines regulation"
In the present study, the authors indicated that CXCL1, CXCL2, 27 CXCL3, and CXCL8 are regulated by E6/E7 of HPV 16 and 18 in addition to be overexpressed in Cervical Carcinoma biopsies, and confirmed that the higher expression of the these chemokines is related to a worse prognostic survival.
The manuscript is well written and the data are represent in best form and the support the main context of the study. In addition , this study will be interest to the reader of the journal. According to the manuscript can be published in the present form.
R: We greatly appreciate your feedback and the time you have spent reviewing our manuscript; thank you very much.

Reviewer 3 Report
This study assessed the prognostic significance of CXCL1-3 and CXCL8 in cervical cancer, exploring their regulation by E6/E7 HPV16 and 18 in SiHa, HeLa, and C33A cell lines. The research is both timely and comprehensive. However, there are some minor points for consideration:
1. In the Introduction, at line 36, it would be beneficial if the authors could include more up-to-date cervical cancer statistics (e.g. data in 2022).
2. In the Introduction, it might be helpful to include a schematic diagram illustrating the relationship between cervical cancer, HPV-16, HPV-18, E6, and E7.
3. Please maintain consistency in using either "HPV16" or "HPV-16" throughout the manuscript. The same consistency should apply to "HPV-18."
4. On page 6, there appears to be a significant amount of unused space in the manuscript that could be optimized.
5. On lines 162-163, consider using "(HPV-)" instead of "(HPVneg)" for clarity.
6. In Section 3.4, specifically on lines 217-218, please provide quantitative data (numerical) explaining why CXCL1, CXCL2, CXCL3, and CXCL8 exhibited high expression.
7. It would be beneficial to include a discussion on the future prospects of this study, such as whether preclinical studies will be continued or other potential avenues for further research.
Author Response
Reviewer 3:
This study assessed the prognostic significance of CXCL1-3 and CXCL8 in cervical cancer, exploring their regulation by E6/E7 HPV16 and 18 in SiHa, HeLa, and C33A cell lines. The research is both timely and comprehensive. However, there are some minor points for consideration:
- In the Introduction, at line 36, it would be beneficial if the authors could include more up-to-date cervical cancer statistics (e.g. data in 2022).
R: Following your recommendation, we cite the most recent worldwide cervical cancer statistics from GLOBOCAN reports in 2020; the cited article was published by Sigh et al. in February 2023 (cite number [2]). In a deeper search, we did not find more recent reports about cervical cancer statistics worldwide. There is only one report by the American Cancer Society (Siegel et al., 2022), which only addresses the incidence and mortality in the U.S.
- In the Introduction, it might be helpful to include a schematic diagram illustrating the relationship between cervical cancer, HPV-16, HPV-18, E6, and E7.
R: We greatly appreciate your suggestion; however, many other review articles address in greater depth the topic of the relationship between cervical cancer and the E6 and E7 oncoproteins of HPV; among them, we can mention Pal A. & Kundu R. 2020; and Hoppe-Seyler K. et al. 2018. In the new version of the manuscript, we have included both references. Therefore, we do not consider it necessary to add a scheme in the introduction section since the topic's breadth exceeds the study's purpose on the modification of chemokine expression in the presence of HPV E6/E7 oncoproteins.
To improve the introduction, we expanded on several topics, including this one.
- Please maintain consistency in using either "HPV16" or "HPV-16" throughout the manuscript. The same consistency should apply to "HPV-18."
R: Thank you very much for the correction. The changes in "HPV-16" and "HPV-18" were already replaced by "HPV16" and "HPV18" every time this term was used—an apology for the inconsistencies and omissions committed.
- On page 6, there appears to be a significant amount of unused space in the manuscript that could be optimized.
R: Thank you very much for the comment; before sending the manuscript, a last-minute modification caused one of the images to be moved down a couple of lines, causing significant blank space. The correction has already been applied in the new version of the manuscript.
- On lines 162-163, consider using "(HPV-)" instead of "(HPVneg)" for clarity.
R: Thank you for your suggestion; the term "HPVneg" has been replaced by "HPV-" every time this term was used.
- In Section 3.4, specifically on lines 217-218, please provide quantitative data (numerical) explaining why CXCL1, CXCL2, CXCL3, and CXCL8 exhibited high expression.
R: In this section, the wording was improved to make the description more straightforward. Additionally, the numerical data of the Hazard ratio was included, as you suggested. It was explained in more detail that the expression profiles were divided into two groups: those patients with CESC who presented high or low expression (considering the median).
- 7.It would be beneficial to include a discussion on the future prospects of this study, such as whether preclinical studies will be continued or other potential avenues for further research.
R: As suggested, we have modified the conclusions section, adding the prospects we have in mind. Thank you for your feedback.

Round 2
Reviewer 1 Report
accepted
Minor editing of English language required
Reviewer 3 Report
I am satisfied with the modifications and corrections made by the authors as per my comments. The quality and presentation of the manuscript are improved.